# Pathogens Associated with Bovine Mastitis: The Experience of Bosnia and Herzegovina

**DOI:** 10.3390/vetsci11020063

**Published:** 2024-02-01

**Authors:** Maid Rifatbegović, Robin A. J. Nicholas, Tarik Mutevelić, Mithat Hadžiomerović, Zinka Maksimović

**Affiliations:** 1Department of Pathobiology and Epidemiology, Veterinary Faculty, University of Sarajevo, Zmaja od Bosne 90, 71000 Sarajevo, Bosnia and Herzegovina; maid.rifatbegovic@vfs.unsa.ba (M.R.); zinka.maksimovic@vfs.unsa.ba (Z.M.); 2The Oaks, Nutshell Lane, Farnham, Surrey GU9 0HG, UK; 3Department of Clinical Sciences, Veterinary Faculty, University of Sarajevo, Zmaja od Bosne 90, 71000 Sarajevo, Bosnia and Herzegovina; tarik.mutevelic@vfs.unsa.ba; 4Cantonal Administration of Civil Protection of Bosnian-Podrinje Canton, 1. Slavne Višegradske Brigade 2a, 73000 Goražde, Bosnia and Herzegovina; direktor_kucz@bpkg.gov.ba

**Keywords:** cattle, clinical and subclinical mastitis, bacteria, mycoplasmas, fungi, algae

## Abstract

**Simple Summary:**

Mastitis is defined as the inflammation of the mammary gland and is one of the most widespread and economically important diseases of dairy cows. Bacteria are the most reported mastitis-causative agents, while other pathogens are often overlooked because they are not routinely investigated. Incomplete diagnosis may result in inappropriate antimicrobial therapy, treatment failure, antimicrobial resistance, dissemination of pathogens, and mastitis recurrences. Thus, this study aimed to investigate the presence of not only bacteria but also other microorganisms associated with cattle mastitis on dairy farms in Bosnia and Herzegovina, a country that lacks an effective mastitis control programme and bacteriological analysis of mastitic milk. The current study revealed *Mycoplasma bovis* as the main pathogen and a variety of other mastitis-causing agents in cattle: bacteria (*Escherichia coli*, *Staphylococcus aureus*, coagulase negative staphylococci, *Streptococcus agalactiae*, *Streptococcus uberis*, and others), fungi (*Candida* spp.), and algae (*Prototheca zopfii*). The finding of mastitis cases requiring currently unavailable treatment and vaccines emerges in the broader scope of etiological agents in routine mastitis diagnosis. These measures applied at the herd and national levels are crucial for more effective mastitis control, animal health and welfare, the dairy industry, and public health.

**Abstract:**

To obtain improved insights into the complex microbial aetiology of bovine mastitis, this study investigated the pathogens involved in cattle mastitis in Bosnia and Herzegovina. A total of 179 milk samples from cows with clinical mastitis (CM) and subclinical mastitis (SCM), as well as eight bulk tank milk (BTM) samples from 48 dairy farms, were analysed by standard bacteriological and mycological methods. Mycoplasma detection and identification were performed using culture techniques and real-time polymerase chain reaction (PCR). A total of 88 (49.2%) mastitis samples were positive for known mastitis pathogens at 32 of 47 farms (68.1%). *Mycoplasma bovis* was a predominant pathogen (25/187; 13.4%) in the majority of herds (14/48; 29.2%) and accounted for 48.9% of positive CM samples. *Escherichia coli* was the second most dominant CM pathogen (34%), followed by *Streptococcus agalactiae* (10.6%), whereas *Staphylococcus aureus* and coagulase-negative staphylococci were the most common in SCM samples (17.1%). Other mastitis pathogens included *Candida* spp. and *Prototheca zopfii*. Two BTM samples were positive for *M. bovis* only, and one was positive for a mixed culture of *S. aureus* and *Streptococcus uberis*. The finding of various causative agents of bovine mastitis, with *M. bovis* emerging as the main pathogen, emphasizes the significance of comprehensive testing that includes not only common mastitis pathogens but also mycoplasmas, fungi, and algae.

## 1. Introduction

Mastitis is defined as the inflammation of the mammary gland and is one of the most widespread and economically important disease of dairy cows [1]. It is significant with respect to animal health and welfare, productive and reproductive performance in the dairy industry, and public health due to zoonotic pathogens, the use of antimicrobials, and the emergence of antimicrobial-resistant bacteria [1,2]. Direct economic losses are related to the costs of treatment, discarded milk, fatalities, repeated cases of mastitis, and veterinary services, while indirect costs involve decreased milk production and quality, increased culling, loss of premiums, pre-term drying off, and other factors [2]. 

Based on the appearance or lack of clinical signs, mastitis can be defined as clinical or subclinical intramammary inflammation [3,4]. Clinical mastitis (CM) is udder inflammation characterized by visible abnormalities in the milk or in the mammary gland [5]. According to severity, it can be classified as mild, moderate, or severe [5,6]. Subclinical mastitis (SCM) is the most prevalent form of mastitis, occurring more often and persisting for longer than CM [3,5]. Contrary to CM, this form of mastitis lacks visible signs in the mammary gland or the milk; thus, it is difficult to detect and may serve as a reservoir for pathogens and their dissemination. SCM results in decreased milk production and increased somatic cell count (SCC) [3] and requires a diagnostic test for detection. The most common test is measurement of SCC [5].

The aetiology of cattle mastitis can be infectious or non-infectious [1,7]. Cell-walled bacteria are the most commonly reported mastitis-causative agents, although a variety of other microorganisms, such as mycoplasmas, chlamydia, algae, fungi, and viruses, have also been associated with the disease [1,7,8]. According to the primary reservoir and mode of transmission, mastitis pathogens have been classified, although not strictly, as contagious or environmental [1,9]. The primary target of infection of contagious pathogens is the mammary gland of infected animals [8]; they are adapted to survive within the udder of the host, where they induce infections and may spread among animals [1]. Environmental or opportunistic invaders are derived from the habitat of the cow rather than from other infected cows [9]. They are non-host-adapted and can be rapidly eliminated by the animal [1]. Infectious agents of bovine mastitis are also divided into major and minor pathogens according to their prevalence and the severity of symptoms [3]. Major contagious pathogens are *Staphylococcus aureus*, *Streptococcus agalactiae*, and *Mycoplasma bovis*, while coagulase-negative staphylococci (CoNS) and *Corynebacterium bovis* are considered minor contagious pathogens. Among numerous environmental pathogens, *Escherichia coli* and *Streptococcus uberis* are the most frequently isolated from mastitis cases [3,7]. A recent global study on bacterial pathogens of bovine mastitis demonstrated that *S. aureus,* followed by CoNS, *E. coli*, *S. agalactiae*, and *S. uberis*, are the major causative agents [10]. Fungi, among which yeasts of the *Candida* genus are most commonly associated with mastitis, are considered minor environmental pathogens [3,8,11]. The incidence of yeast mastitis has increased in recent years, and the frequent isolation of *Candida* spp. or outbreaks of *Candida* mastitis has been reported in several countries [7,11,12]. For algae of the genus *Prototheca*, it is unclear whether they are contagious or environmental pathogens [9]; nevertheless, an increasing number of mastitis cases has been observed in recent decades [13,14,15]. 

The prevalence of the different pathogens varies between countries or even herds. Causative agents previously considered minor pathogens have replaced previously important pathogens [1,3]. 

Accurate and prompt detection of mastitis-causing agents is essential for the diagnosis of intramammary infection, effective treatment, prevention of recurrent infections, and improvements of control measures [1]. Control is complicated by the fact that the aetiology of mastitis is not limited to a single pathogen; multiple agents can be implicated concurrently or subsequently, resulting in polymicrobial disease. However, most previous studies have focused on bacteria involved in cattle mastitis [3,10], leaving a gap in terms of the true prevalence and significance of other pathogens. Mastitis-causing agents such as mycoplasmas are often overlooked because they are not routinely identified, despite the recent impact of *M. bovis* infections on the dairy sector [16,17,18]. 

Traditionally, Bosnia and Herzegovina (B&H) is characterised by smallholder farming; however, dairy farms have been consolidating lately, with the closure of small farms and the expansion of large farms. The number of cattle is estimated at 332,000, of which 250,000 are dairy cows. Cow milk dominates total milk production, with a 96% share [19]. The common breeds of cows are imported, mainly Simmental, Holstein Friesian, and crossbreeds. B&H still lacks veterinary herd health management (VHHM) programs and an effective mastitis control programme. The programme of animal health protection measures and their implementation for 2023, based on the provisions of the Veterinary Law in B&H and implemented by the Veterinary Office of B&H, regulates requirements for udder health control, which include the plate count and somatic cell count of milk intended for public consumption. Bacteriological analysis of mastitic milk has not been mandatory since 2014 due to the lack of legislative requirements; samples are not usually submitted for testing until mastitis has been observed and/or antimicrobial treatment has failed [20]. 

Data on bovine mastitis-causing agents in B&H are scarce [21,22], and conducted studies have reported *S. aureus* as the most frequently isolated bacterial pathogen. Thus, due to the need for a large-scale study to obtain improved insights into the complex microbial aetiology of cattle mastitis, this study aimed to investigate the presence and frequency of bacteria, mycoplasmas, fungi, and algae associated with CM and SCM in dairy farms in B&H.

## 2. Materials and Methods

### 2.1. Study Area, Farms, and Sample Collection 

B&H is located in the western Balkan Peninsula of southeastern Europe and covers an area of 51.209.2 km^2^. The climate of B&H varies from a temperate continental climate in the northern and central parts to an alpine climate in the mountain regions and a Mediterranean climate in the south. The average annual precipitation is about 1.250 mm. The average annual temperature in the lowland area of northern B&H ranges approximately between 10 °C and 12 °C and between 12 °C and 17 °C in the coastal area.

This study was conducted on 48 dairy cattle farms in different regions of B&H in the period between October 2018 and February 2022. The farms were classified into four herd size categories based on the number of cows: individual (≤5 cows), small (6–20 cows), medium (21–49 cows), and large (≥50 cows). A total of 179 milk samples were collected from lactating cows with CM (*n* = 68) and SCM (*n* = 111) from dairy farms (*n* = 47) located in the following 22 municipalities: Cazin, Bihać, Prnjavor, Doboj South, Gradačac, Pelagićevo, Bijeljina, Kalesija, Vitez, Kakanj, Visoko, llijaš, Hadžići, Ilidža, Han Pijesak, Sokolac, Rogatica, Goražde, Rudo, Foča, Gacko, and Čapljina (Figure 1).

SCM was defined as a high somatic cell count (SCC) detected by direct measurement and/or by the California mastitis test (CMT) without clinical signs. Additionally, bulk tank milk (BTM) samples (*n* = 8) were obtained from six of these farms, including four large (*n* = 5) farms (two BTM samples were collected from the same large farm) and two medium (*n* = 2) farms, as well as from one individual (*n* = 1) farm from which the samples were not collected individually (Appendix A).

All samples were collected aseptically in sterile 50 mL containers and transported in cooler boxes with ice packs to the microbiology laboratory of the Department of Pathobiology and Epidemiology, Veterinary Faculty, University of Sarajevo. 

### 2.2. Bacteriological Analysis

Milk samples were analysed by standard bacteriological methods [8]. Briefly, 100 μL of each sample was inoculated onto blood agar (with 7% sheep blood), bromocresol purple lactose agar, and MacConkey agar (Condalab, Madrid, Spain). The plates were incubated aerobically at 37 °C and inspected after 24, 48, and 72 h. The pure culture of the isolates was obtained by subculturing of a single well-isolated colony. The isolates were identified by cultural (colony morphology, haemolysis, and lactose fermentation), microscopic (Gram staining, cell morphology, and motility), and biochemical examination. Biochemical tests included oxidase, L-pyrrolidonyl-β-naphthylamide (PYR) (TestLine, Brno, Czech Republic), coagulase (BD, Sparks, MD, USA), catalase, urease, indole, citrate, maltose, esculin, nitrate, triple sugar iron agar (TSI), oxidative-fermentative (OF), *O*-nitrophenyl-β-D-galactopyranoside test (*β*-galactosidase), sensitivity to polymyxin B, bacitracin susceptibility (0.04 units) (Condalab, Madrid, Spain), and CAMP. Milk samples that had three or more dissimilar colonies, with the absence of the pathogens and predominant colony type, were regarded as contaminated and rejected.

### 2.3. M. bovis Detection and Identification

#### 2.3.1. Isolation

For isolation of mycoplasmas, a 200 µL aliquot of each milk sample was serial-diluted in Thiaucourt’s liquid medium (2 mL) [23], and a few drops of the samples were directly spread onto the same solid medium. Simultaneously the samples were filtered through a 0.45 μm membrane filter (Lab Logistics Group, Meckenheim, Germany) into a broth and plate. The broths were incubated aerobically at 37 °C for 3 days and monitored daily, and those with suspected growth were subcultured in liquid and onto solid media. A blind passage from broths with no change was performed after 48 and 72 h of incubation. All plates were incubated in a 95% N_2_ and 5% CO_2_ humidified atmosphere at 37 °C for seven days and examined every second day under a stereo microscope for typical mycoplasma ‘fried egg’ colonies and, eventually, film and spot production [24]. 

#### 2.3.2. Real-Time PCR Detection and Identification

Real-time PCR specific for *M. bovis* was performed on DNA extracts from broth cultures and milk samples. 

##### DNA Extraction from the Isolates

A 4 mL aliquot of a broth culture was centrifuged at 10,000 rpm for 10 min, and the pellet was washed and resuspended in 100 µL of TE buffer (10 mM Tris-HCl, 1 mM EDTA, pH 8). The suspension was then heated in a dry block at 100 °C for 15 min, chilled for 5 min, and centrifuged at 13,200 rpm for 5 min. The supernatant containing DNA was collected and kept at −20 °C until testing.

##### DNA Extraction from Milk Samples

A 500 µL aliquot of a milk sample was centrifuged at 5200 rpm for 10 min, and the pellet was suspended in 180 µL of the tissue lysis buffer (ATL) and 20 µL of proteinase K provided in the DNeasy Blood & Tissue Kit (Qiagen, Hilden, Germany). DNA was extracted following the manufacturer’s instructions, eluted with 100 μL of AE buffer, and stored at −20 °C until use.

##### *M. bovis* Real Time PCR

A real-time PCR assay targeting the 3′-terminal region of the *opp*D gene was used for the detection of *M. bovis* as described previously [25]. The reaction mixes were made up in 25 μL volumes containing 12.5 μL Luna^®^ Universal qPCR Master Mix (New England Biolabs), primers PMB996-F (5′-TCAAGGAACCCCACCAGAT-3′) and PMB1066-R (5′-AGGCAAAGTCATTTCTAGGTGCAA-3′), and the Mbovis1016 probe (FAM-TGGCAAACTTACCTATCGGTGACCCT-TAMRA) (Eurofins, MWG, Operon) at concentrations of 0.3 μM and 0.2 μM, respectively, and 2.5 μL extracted DNA. A positive template control (DNA from *M. bovis* PG45 strain) and a negative control (nuclease-free water) were included in all assays. Analysis was performed in a Stratagene Mx3005P qPCR System (Agilent Technologies, United States) with an initial denaturation at 95 °C for 10 min, followed by 45 cycles of denaturation at 95 °C for 30 s and annealing and extension at 60 °C for 60 s. 

### 2.4. Isolation and Identification of Yeasts and Algae 

The samples were inoculated in duplicate on blood agar containing 7% sheep blood and Sabouraud chloramphenicol dextrose agar (SDA) (Condalab, Madrid, Spain) and incubated aerobically at 37 °C for 24–72 h and at 25 °C for seven days. The yeasts and algae were identified using colony morphology and microscopic examination [8,14].

### 2.5. Statistical Analysis 

Statistical evaluations were performed by Fisher’s exact test and the chi-square test. A *p* value ≤ 0.05 was considered statistically significant. 

## 3. Results

### 3.1. Presence of Pathogens

A total of 88 (49.2%) mastitis samples were positive for one or multiple pathogens and detected at 32 of 47 farms (68.1%) (Table 1, Table 2 and Table 3). Microorganisms were more often found in cattle with CM (47/68; 69.1%) compared to SCM cases (41/111; 36.9%) (*p* < 0.001). A sole pathogen was significantly more frequent (73/88; 83%) than multiple pathogens (15/88; 17%) (*p* < 0.001) (Table 2 and Table 3). 

#### 3.1.1. Bacteria

Bacteria were isolated from 71 samples (39.7%), among which *E. coli* was the most commonly detected (*n* = 20; 28.2%), followed by *S. aureus* (*n* = 10; 14.1%) (Table 3). Following *M. bovis*, *E. coli* was the second most frequently identified pathogen in CM cases (16/47; 34%), followed by *S. agalactiae* (5/47; 10.6%). *E. coli* was significantly more common in CM (16/68; 23.5%) than in SCM samples (4/111; 3.6%) (*p* < 0.001). The most common bacteria isolated from SCM cases were *S. aureus* and CoNS (7/41; 17.1%), followed by *S. uberis* and *Streptococcus* spp. (5/41; 12.2%), respectively (Table 3). CoNS were isolated only from SCM cases (7/111; 6.3%) (Figure 2 and Figure 3). 

#### 3.1.2. *Mycoplasma bovis*

*M. bovis* was a predominant pathogen (25/187; 13.4%), accounting for 26.1% of positive samples (*n* = 88). It was isolated from 7 of 23 *M. bovis* PCR-positive milk samples (30.4%), all from cattle with CM (33.8%), and accounted for 48.9% of positive CM samples. All SCM cases were negative for *M. bovis*. *M. bovis* was found alone (13/23; 56.52%) and with other pathogens (10/23; 43.48%) as a predominant pathogen in polymicrobial samples (10/15; 66.6%) (Table 3 and Table 4). 

#### 3.1.3. Yeasts and Algae

*Candida* spp. Was isolated from three samples (3.4%), of which two were positive on bacteria or *M. bovis* (Table 4). *P. zopfii* was found in two samples (2.3%) in pure culture. 

#### 3.1.4. Bulk Tank Milk Results

A total of two (25%) BTM samples were only positive for *M. bovis* detected by real-time PCR, whereas a mixed bacterial culture of *S. aureus* and *S. uberis* was isolated from one sample (12.5%). These positive samples were obtained from two large farms. Other BTM samples were negative for all pathogens (5/8; 62.5%). 

#### 3.1.5. Distribution of Mastitis Pathogens

*M. bovis* was the most common pathogen in the majority of herds (14/48; 29.2%), significantly more detected in smaller herds (≤49 cows) than in larger herds (≥50 cows) (*p* = 0.014) (Figure 2 and Figure 3). The next most common was *E. coli* (10/48; 20.8%), followed by *S. aureus* (7/48; 14.6%), CoNS and *Streptococcus* spp., with 12.5% recorded, respectively (6/48), and *S. uberis* (4/48; 8.3%) (Table 3 and Appendix A). 

## 4. Discussion

This study revealed the presence of various mastitis-causing agents including *M. bovis* as the main mastitic pathogen in dairy herds in B&H. Mycoplasmas have been rarely investigated in undiagnosed cases of mastitis, estimated at over a quarter of clinical and nearly 40% of subclinical cases [16]. Among bovine mastitis-associated mycoplasmas, *M. bovis* is the predominant causative agent worldwide [18,26,27]. The prevalence of *M. bovis* mastitis varies globally, as well as between and within herds, and changes over time [16,27]. This emerging pathogen of dairy cattle [26] occurs as a cause of endemic subclinical disease and CM outbreaks [16]. Severe outbreaks have been reported across the globe. They usually arise when *M. bovis* is introduced into disease-free areas [27]. The detection of *M. bovis*-positive herds in B&H in various locations has exposed the wide dissemination of this contagious agent in the country. *M. bovis* was not previously detected in milk samples, despite its frequent isolation from the respiratory tract of cattle in B&H [21]. Under stress conditions, mycoplasmas present in the respiratory system or other body sites of animals showing no clinical signs may enter into the mammary gland and produce CM [28]. However, prior presence of *M. bovis* in milk samples cannot be excluded, mainly because mycoplasmas are not routinely investigated. The current finding and a source of *M. bovis* mastitis could be related to the frequent introduction of new animals of unknown health status into the country/farms, resulting in consequent spread, as was previously reported in other countries [26,27]. Imported bovine semen is commonly used for the insemination of cattle in B&H and, uncontrolled for the presence of this mycoplasma, may serve as a source of infection [27,29]. 

The relationship between herd sizes and the risk of mycoplasmal infection is unclear. Large herds seem to be at a higher risk, most likely due to the frequent purchase of cattle and the greater opportunity for *M. bovis* to spread and persist [16,26]. Still, infections also occur in small herds [28,29]. In this study *M. bovis* was more common in smaller herds (≤49 of cows) than in larger herds, and the reason may lie in smallholders often purchasing cows with lower productivity or high SCC from large farms, lacking quarantine or other prevention and control measures. 

*M. bovis* CM occurred in 25.5% of the herds and varied between them. This variation should be interpreted with caution, considering the limitation in terms of the small number of samples, ranging from 1 to 38 per herd. The number of cases of *M. bovis* CM (33.8%) appears to be higher than those reported in other European countries (11%) [30,31] and lower than in Australia (76%) [32] and the USA (85%) [26]. However, these differences may be related to the variable number of samples in each study and other factors such as control of introduced animals, mastitis control programmes and treatment, and herd/farm size and management practice. A decline in *M. bovis* mastitis cases within herds was observed in countries that monitored herd health status following outbreaks or reported high prevalence of *M. bovis* [29,33]. This reduced number can be attributed to effective control measures; however, the possibility of spontaneous clearance of the infection within months of outbreaks cannot be completely ruled out [16,28]. Nevertheless, clinically healthy cows, having already experienced mastitis, become a permanent reservoir of the pathogen; thus, *M. bovis* may remain in previously affected herds and persist in the environment for months [28]. In the present study, following the detection of *M. bovis*-positive clinical cases, diseased cows were slaughtered or culled from the herds of most large farms to prevent transmission, while the measures conducted on smaller farms are unknown. More effective control includes more frequent testing of herds after disease occurrence, since subclinical intramammary infections can occur as a consequence of CM and, therefore, enable *M. bovis* persistence and spread in the herd. This should be achieved by individual testing, considering that whole-herd sampling after CM outbreaks is not indicated for the detection of subclinical *M. bovis* infection [27,34]. 

Mycoplasma SCM cases are relatively underestimated [26], mostly because of the lack of routine mycoplasma investigations [16]. Conducted studies have reported low apparent *M. bovis* prevalence at the cow level (0–0.2%) [34] and a within-herd prevalence of 17.2% [35]. In the present study, the absence of *M. bovis* in SCM cases could be explained by low-level or intermittent shedding of mycoplasmas for variable periods, frequently reported with chronic and mycoplasma SCM cases. Thus, multiple testing of apparently negative samples should be applied to overcome false-negative results [28]. 

Co-infections in *M. bovis*-associated mastitis are underinvestigated, and their effect on the severity of mastitis is uncertain. In the current study, *M. bovis* was found in CM samples alone and in combination with other pathogens. Although rarely reported in the UK [30], 71% of *M. bovis*-associated mastitis diagnoses involved just *M. bovis*. In contrast, the majority of cows (57%) with CM in Australia had *M. bovis* combined with various cell-walled bacteria [32]. 

The global prevalence of major mastitis-causing bacteria *S. aureus*, CoNS, *E. coli*, *S. agalactiae*, and *S. uberis* was estimated at 25, 20, 11, 9, and 9%, respectively [10]. In this study, *E. coli* emerged as the second most dominant pathogen, more frequently detected in CM cases than in SCM cases. Although considered a main cause of acute CM, *E. coli* can also be involved in SCM cases [9]. Contrary to *S. uberis*, *S. agalactiae* was recovered more often from clinically infected cows than subclinically infected cows. Associated with CM, both streptococcal species usually induce SCM [3,10]. The finding of *S. aureus* and CoNS as the most common pathogens in SCM samples agrees with their comprehensive status [10]. In contrast to the present study, in a previous study conducted in the Zenica region of B&H [22], *S. aureus* (21.7%) and *S. agalactiae* (17.4%) were the most commonly identified isolates among the bacteria recovered from 23 of 52 CM samples. The other isolates included CoNS and *Klebsiella pneumoniae* (13.04%), followed by *Enterococcus* spp. and *E. coli* (8.7%), while *Streptococcus* spp., *Enterobacter* spp., *Serratia* spp., and *Yersinia enterocolitica* were less frequently detected [22].

Like mycoplasmas, algae are often neglected mastitis pathogens. *Prototheca* mastitis has recently become an emerging disease [13,14,15]. This alga mainly causes chronic, symptom-less mastitis, although acute CM also occurs [8,13]. Despite the low numbers, *P. zopfii* was detected for the first time in B&H, being isolated in pure culture from high-yielding dairy cows with CM and SCM on two large farms. *Prototheca* mastitis may occur in well-managed herds, and its greater presence appears to be correlated with larger herds [13]. 

*Candida* spp. was found in milk samples from cows with CM and SCM. Prolonged or repetitive use of antibiotics contributes to fungal intramammary infections, with a tendency toward a chronic form resistant to treatment [8]. Although isolated from CM and SCM cases and milk samples from healthy cows, several countries have reported a dominance of *Candida* in CM samples [8,11,12].

Polymicrobial mastitis raises the question of whether *M. bovis* contributes to intramammary infections (more severe or persistent mastitis) by other pathogens or the other way around, as their concurrent or consecutive occurrence, as well their synergistic effect remains undisclosed. Nevertheless, multiple infections combined with the increasing antimicrobial resistance of the pathogens, a very difficult-to-treat or non-treatable mastitis, long-term survival of pathogens by forming biofilms, and a lack of effective vaccines underline difficulties in therapy and controlling bovine mastitis. 

Finally, in addition to the extra testing required, the results underline the need for improved farm management practices, including selecting out susceptible dairy cattle based on risk factors such as parity, age, and lactation.

## 5. Conclusions

The current study identified *M. bovis* as the main pathogen on dairy farms, as well as a variety of other bovine mastitis-causing agents in a country that lacks a mastitis control programme and bacteriological diagnostics for mastitic milk samples. 

Since there are no effective treatments or vaccines for bovine mastitis, which can be caused by a range/combination of microbial pathogens, it is crucial that regular testing of mastitic samples for all these organisms be carried out at the herd and national levels.

## Figures and Tables

**Figure 1 vetsci-11-00063-f001:**
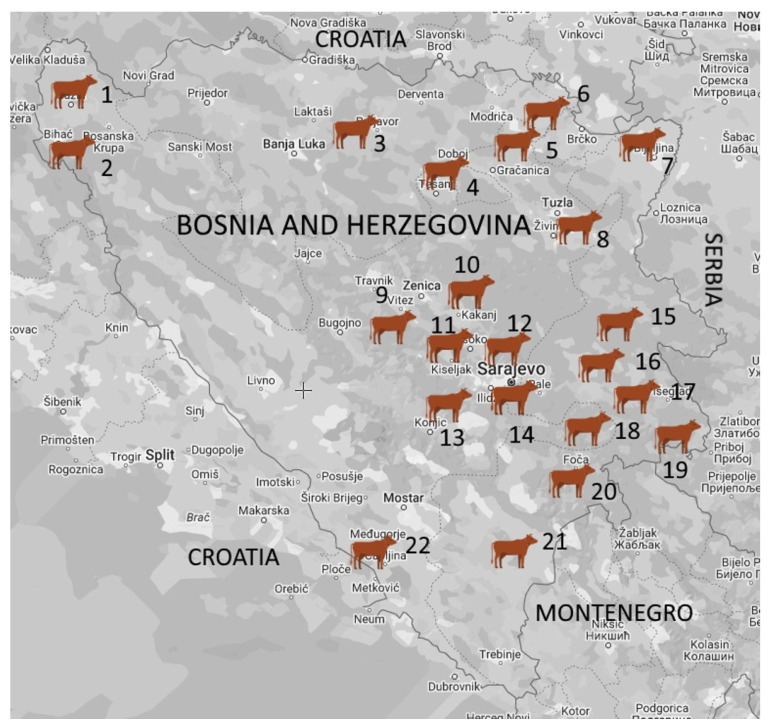
Geographic distribution of the tested dairy herds in Bosnia and Herzegovina. The number of dairy farms and samples from lactating cows are listed next to the name of the municipality: 1. Cazin (1/11); 2. Bihać (1/9); 3. Prnjavor (1/1); 4. Doboj South (1/5); 5. Gradačac (1/10); 6. Pelagićevo (1/10); 7. Bijeljina (6/11); 8. Kalesija (1/10); 9. Vitez (1/38); 10. Kakanj (4/5); 11. Visoko (1/1); 12. llijaš (2/4); 13. Hadžići (1/2); 14. Ilidža (2/11); 15. Han Pijesak (12/18); 16. Sokolac (2/6); 17. Rogatica (1/1); 18. Goražde (1/2); 19. Rudo (3/4); 20. Foča (1/1); 21. Gacko (2/2); 22. Čapljina (1/17).

**Figure 2 vetsci-11-00063-f002:**
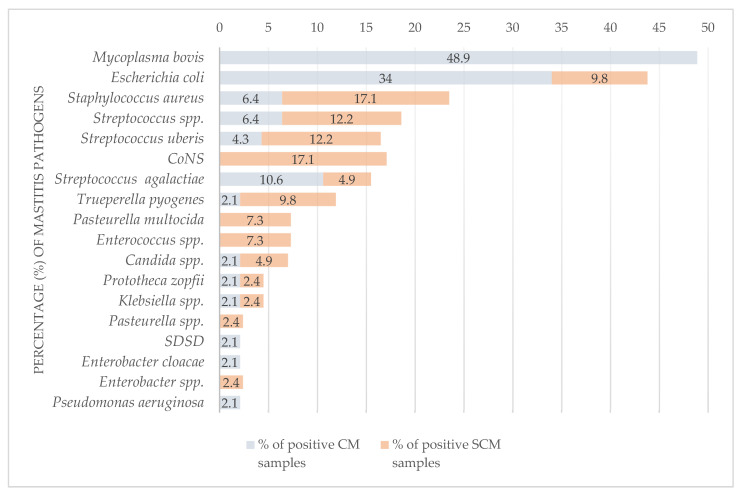
Comparison of pathogens detected in milk samples from cattle with clinical and subclinical mastitis. CM—clinical mastitis; SCM—subclinical mastitis; M. bovis—culture-positive and direct real-time PCR-positive milk samples; CoNS—coagulase-negative staphylococci; SDSD—Streptococcus dysgalactiae subsp. dysgalactiae. The results of testing of the BTM samples are excluded.

**Figure 3 vetsci-11-00063-f003:**
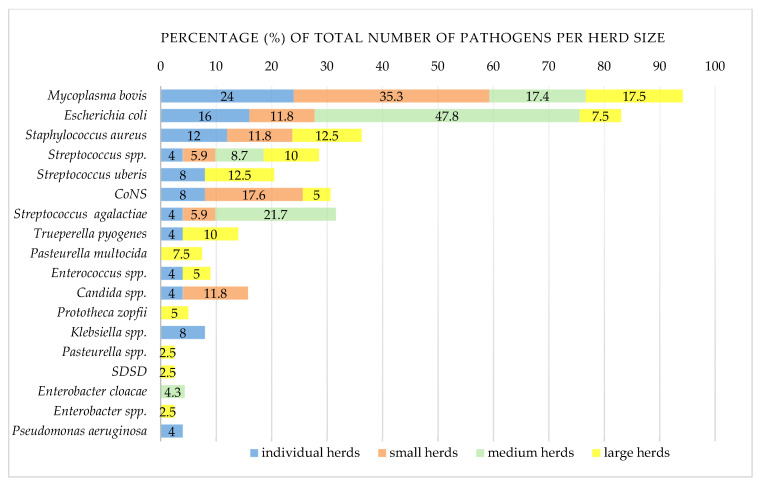
Comparison of clinical and subclinical mastitis pathogens in dairy cows according to herd size. Number of positive/tested herds: individual herds (≤5 cows), 14/19; small herds (6–20 cows), 9/17; medium herds (21–49 cows), 2/3; large herds (≥50 cows), 7/8*M. bovis*—culture-positive and direct real-time PCR-positive milk samples; CoNS—coagulase-negative staphylococci; SDSD-*Streptococcus dysgalactiae* subsp. *dysgalactiae*. The results of testing of the BTM samples were excluded.

**Table 1 vetsci-11-00063-t001:** Pathogen-positive milk samples detected in cattle with mastitis.

Farm Size	No. of Positive Farms (%)	Clinical Mastitis		Subclinical Mastitis		Total	
		No. of Samples	No. of Positive Samples (%)	No. of Samples	No. of Positive Samples (%)	No. of Samples	No. of Positive Samples (%)
Individual (≤5 cows)	14/19 (73.7)	17	12 (70.6)	7	6 (85.7)	24	18 (75)
Small (6–20 cows)	9/17 (52.9)	10	7 (70)	18	6 (33.3)	28	13 (46.4)
Medium (21–49 cows)	2/3 (66.7)	21	18 (85.7)	9	0	30	18 (60)
Large (≥50 cows)	7/8 (87.5)	20	10 (50)	77	29 (37.7)	97	39 (40.2)
Total (%)	32/47 (68.1)	68	47 (69.1)	111	41 (36.9)	179	88 (49.2)

**Table 2 vetsci-11-00063-t002:** Monomicrobial and polymicrobial samples from clinical and subclinical mastitis cases.

	**Total (%)**		**Monomicrobial Samples**		**Polymicrobial Samples**	
	No. of Samples (%)	No. of Positive Samples (%)	No. of Positive Samples (%)	% of Total Samples	No. of Positive Samples (%)	% of Total Samples
Clinical mastitis	68 (38)	47 (69.1)	35 (74.5)	51.5	12 (25.5)	17.6
Subclinical mastitis	111 (62)	41 (36.9)	38 (92.7)	34.2	3 (7.3)	2.7
Total (%)	179	88 (49.2)	73 (83)	40.8	15 (17)	8.4

**Table 3 vetsci-11-00063-t003:** Pathogens detected in milk samples from cattle with clinical and subclinical mastitis.

Pathogen						Monomicrobial		Polymicrobial		Clinical Mastitis		Subclinical Mastitis	
	No. of Isolates	% of Total Samples	% of Positive Samples	% of Isolates	No. of Herds (%)	No. of Isolates (%)	% of Positive Samples	No. of Isolates (%)	% of Positive Samples	No. of Isolates (%)	% of Positive Samples	No. of Isolates (%)	% of Positive Samples
*M. bovis* ^1^	23	12.8	26.1	21.9	12 (25.5)	13 (12.4)	17.8	10 (9.5)	66.7	23 (21.9)	48.9	0	0
*E. coli* ^2^	20	11.2	22.7	19	10 (21.3)	13 (12.4)	17.8	7 (6.7)	46.7	16 (15.2)	34	4 (3.8)	9.8
*S. aureus*	10	5.6	11.4	9.5	6 (12.8)	8 (7.6)	11	2 (1.9)	13.3	3 (2.9)	6.4	7 (6.7)	17.1
*Streptococcus* spp.	8	4.5	9.1	7.6	6 (12.8)	5 (4.8)	6.8	3 (2.9)	20	3 (2.9)	6.4	5 (4.8)	12.2
*S. uberis*	7	3.9	8	6.7	3 (6.4)	7 (6.7)	9.6	0	0	2 (1.9)	4.3	5 (4.8)	12.2
CoNS ^3^	7	3.9	8	6.7	6 (12.8)	7 (6.7)	9.6	0	0	0	0	7 (6.7)	17.1
*S. agalactiae*	7	3.9	8	6.7	3 (6.4)	4 (3.8)	5.5	3 (2.9)	20	5 (4.8)	10.6	2 (1.9)	4.9
*Trueperella pyogenes*	5	2.8	5.7	4.8	3 (6.4)	3 (2.9)	4.1	2 (1.9)	13.3	1	2.1	4 (3.8)	9.8
*Pasteurella multocida*	3	1.7	3.4	2.9	2 (4.3)	3 (2.9)	4.1	0	0	0	0	3 (2.9)	7.3
*Enterococcus* spp.	3	1.7	3.4	2.9	2 (4.3)	2 (1.9)	2.7	1	6.7	0	0	3 (2.9)	7.3
*Candida* spp.	3	1.7	3.4	2.9	3 (6.4)	1	1.4	2 (1.9)	13.3	1	2.1	2 (1.9)	4.9
*P. zopfii*	2	1.1	2.3	1.9	2 (4.3)	2 (1.9)	2.7	0	0	1	2.1	1	2.4
*Klebsiella* spp.	2	1.1	2.3	1.9	2 (4.3)	1	1.4	1	6.7	1	2.1	1	2.4
*Pasteurella* spp.	1	0.6	1.1	1	1 (2.1)	0	0	1	6.7	0	0	1	2.4
SDSD ^4^	1	0.6	1.1	1	1 (2.1)	1	1.4	0	0	1	2.1	0	0
*Enterobacter cloacae*	1	0.6	1.1	1	1 (2.1)	1	1.4	0	0	1	2.1	0	0
*Enterobacter* spp.	1	0.6	1.1	1	1 (2.1)	1	1.4	0	0	0	0	1	2.4
*Pseudomonas aeruginosa*	1	0.6	1.1	1	1 (2.1)	1	1.4	0	0	1	2.1	0	0
Total (%)	105	58.7	-	100	-	73 (69.5)		32 (30.5)	-	59 (56.2)	-	46 (43.8)	-

^1^ *M. bovis*—culture-positive and direct real-time PCR-positive milk samples; ^2^ *E. coli*—10 isolates were β haemolytic, and 10 were non-haemolytic; ^3^ CoNS—coagulase negative staphylococci; ^4^ SDSD—*Streptococcus dysgalactiae* subsp. *dysgalactiae*. The results of testing of the BTM samples are excluded.

**Table 4 vetsci-11-00063-t004:** Mastitis samples with more than one detected pathogen.

Pathogens	No. of Positive Samples
*Mycoplasma bovis*/*Escherichia coli*	3
*Mycoplasma bovis*/*Streptococcus agalactiae*	3
*Escherichia coli*/*Staphylococcus aureus* ^1^	2
*Mycoplasma bovis*/*Streptococcus* spp.	2
*Mycoplasma bovis*/*Candida* spp.	1
*Mycoplasma bovis*/*Trueperella pyogenes*	1
*Escherichia coli*/*Streptococcus* spp.	1
*Trueperella pyogenes*/*Pasteurella* spp. ^2^	1
*Escherichia coli*/*Klebsiella* spp./*Enterococcus* spp./*Candida* spp. ^2^	1
Total	15

^1^ Isolates from clinical and subclinical mastitis samples. ^2^ Isolates from subclinical mastitis samples. Other pathogens were detected in clinical mastitis cases.

## Data Availability

Data contained within the article and Appendix A.

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
