# Peer review of "Pathogens Associated with Bovine Mastitis: The Experience of Bosnia and Herzegovina"

_vetsci, 2024, doi:10.3390/vetsci11020063_

Round 1

Reviewer 1 Report

Comments and Suggestions for Authors

This well written report by Rifatbegovic et al., has characterised 179 milk samples from cows with clinical and subclinical mastitis from 48 farms throughout Bosnia and Herzegovina. Using comprehensive isolation techniques, the group has identified Mycoplasma bovis as being the most prevalent bacterial agent of mastitis in this country. This result was not necessarily expected but is well described in the text. 

This work gives a thorough account of the pathogens associated with bovine mastitis in Bosnia and Herzegovina and I have no specific recommendations on how to improve the work or text. 

Comments on the Quality of English Language

Overall the English is very good. Minor edits to ensure correct inclusion of 'a/an' and 'the' is all that is needed.

Author Response

Corrections to grammar have been made

Reviewer 2 Report

Comments and Suggestions for Authors

Dear authors,

The present document is related with one of the hardest pathogen that ocasionally affected dairy cows, but is an important healthy problem.

You can find several comments into the document to analyzed to improve them.

Author Response

Author response: These data are correct; a total of 179 samples from lactating cows were obtained from 47 dairy farms, while BTM samples (n=8) were obtained from seven farms, including one individual (n=1) farm from which milk samples were NOT individually collected, two medium (n=2) and four large farms (n=5) (two BTM samples were collected from the same large farm). Apart from BTM samples CM and/or SCM samples were also collected from these medium and large farms. To make this clear (179 of CM and SCM milk samples from 47 dairy farms; 8 BTM samples from 6 of these farms and from an additional farm-7 farms; total 48 dairy farms and 187 samples) we have added additional data in the manuscript, including the data in Table S1.

Page 3

lines 137-138: Please, check the quantity of samples (1+2+4=7) not 8

Author response: Considering that two BTM samples were collected from the same large the quantity of samples is: 1+2+5=8. We have added the information: “Additionally, bulk tank milk (BTM) samples (n=8) were obtained from six of these farms, including four large (n=5) farms (two BTM samples were collected from the same large farm) and two medium (n=2) farms, and from one individual (n=1) farm from which the samples were not collected individually (Table S1).”

line 139: Check the correct abreviation (mL)

Author response: In accordance with your recommendation, we have corrected the abbreviation (ml to mL).

line 141

Author response: We have corrected faculty to Faculty.

Page 4

lines 150-154: Please, add a description of this data into the M&M section.

Author response: In accordance with your recommendation, we have added the data in the M&M section: “A total of 179 milk samples were collected from lactating cows with CM (n=68) and SCM (n=111) from dairy farms (n=47) located in 22 following municipalities: Cazin, Bihać, Prnjavor, Doboj South, Gradačac, Pelagićevo, Bijeljina, Kalesija, Vitez, Kakanj, Visoko, llijaš, Hadžići, Ilidža, Han Pijesak, Sokolac, Rogatica, Goražde, Rudo, Foča, Gacko and Čapljina (Figure 1).”    

line 150

Author response: We have corrected caws to cows.

Page 5

line 180

Author response: This gas mixture (95% N2 and 5% CO2) was used for the isolation of mycoplasmas. Not sure how to correct.

lines 191 and 199

Author response: We have corrected the abbreviation (ml to mL).

lines 209 and 213

Author response: We have corrected the abbreviation (µl to µL).

Page 6

line 232: What happened with farms #48...???? Please, change by Table 1-3.

Author response: In this sentence “A total of 88 (49.2%) mastitis samples were positive for one or multiple pathogens and detected at 32 of 47 farms (68.1%) (Table 1-3), we describe the results of testing of mastitis samples (CM and SCM), and these samples were collected from 47 farms (one BTM sample was obtained from one individual farm).

Author response: In accordance with your instruction, we have changed Table 1, Table 2 and Table 3 by Table 1-3.

line 235: Change by Table 2 and 3

Author response: In accordance with your instruction, we have changed Table 2 and Table 3 by Table 2 and 3.

line 237: Please, add into M&M section a description of criteria of type and quantity of pathogen isolated to considered as non-contaminated.

Author response: We have described the criteria in the M&M section: “Milk samples that had three or more dissimilar colonies, with the absence of the pathogens and predominant colony type, were regarded as contaminated and rejected”.

lines 239-241: Please, re-edited the text to get more clear for readers.

Author response: We have edited the text to be clearer.

line 239: In the section 3.1 says 88 microbiological positive samples...??????

Author response: As stated, a total of 88 mastitis samples were positive for one or multiple pathogens, while the bacteria only were isolated from 71 samples.

line 252: Change by "Table 3 and 4"

Author response: In accordance with your instruction, we have changed Table 3 and Table 4 by Table 3 and 4.

Page 7

Table 3. It is much better to describe complete name for first time.

Author response: According your instruction, we have used complete bacterial names (Trueperella pyogenes, Pasteurella multocida, Enterobacter cloacae, Pseudomonas aeruginosa) in Table 3.

Page 9

line 338

Author response: Following your corrections we have deleted all “%” apart from the last one.

Page 10

line 367-371: It is clear tht there is a big sanitary problem around of dairy herds in the country, particularly with M. bovis. I think that previous to suggested a protocol of vaccines or treatments, it will be a dairy cattle analysis census by number of parity, age and lactation to take decisions to eliminated in the future the positive animals  

Author response: We have included a modification of the reviewers suggestion and placed it at the end of the discussion

Reviewer 3 Report

Comments and Suggestions for Authors

I would like to express my gratitude for offering me the opportunity to review the paper. In this study, pathogens associated with bovine mastitis in Bosnia and Herzegovina have been investigated, and this study aimed to investigate the presence not only bacteria but also other microorganisms associated with cattle mastitis on dairy farms in Bosnia and Herzegovina, a country which lacks effective mastitis control program and bacteriological analysis of milk.

Although there are not many technical innovations in this research work, this study has clinical significance for drawing up an effective mastitis control program in B&H.

I do, however, have some suggestions for improvement of the manuscript. I think this manuscript needs major revision before being considered for acceptance.

1, in the “introduction” section, some literature published recently (within 3 years) needs to be added.

2, The results of the study need to be further analyzed and compared, e.g. species differences in bacteria isolated from CM, SCM, and BTM? species differences between regions; species differences in bacteria isolated from different farm sizes?

In other words, the data analysis needs to be further enriched, and some graphs should be used for presentation.

 3, p273, in the “discussion” section, please add some study results published in B & H, and compare your results with other literature published before.   

Comments on the Quality of English Language

Moderate editing of English language required

Author Response

1, in the “introduction” section, some literature published recently (within 3 years) needs to be added.

Author response: According your recommendation, we have added some literature published recently in the “introduction” section.

2, The results of the study need to be further analyzed and compared, e.g. species differences in bacteria isolated from CM, SCM, and BTM? species differences between regions; species differences in bacteria isolated from different farm sizes?

In other words, the data analysis needs to be further enriched, and some graphs should be used for presentation.

Author response: Precise description of the data is provided in the section “Results” including Supplementary table 1 (Table S1). In accordance with your instructions, we have compared the results and presented them in additional figures-Figure 2 and Figure 3.

3, p273, in the “discussion” section, please add some study results published in B & H, and compare your results with other literature published before.   

Author response: Studies on mastitis pathogens in dairy cattle in B&H are very limited, as described in sections “Introduction” (“Data on bovine mastitis-causing agents in B&H are scarce [21,22], and the conducted studies reported S. aureus as the most frequently isolated bacterial pathogen.”) and “Discussion” („M. bovis was not previously detected in milk samples, despite its frequent isolation from the respiratory tract of cattle in B&H [21].”). According to your recommendation, we have added the results of a previous study in the “discussion” section.

Comments on the Quality of English Language: Moderate editing of English language required

Author response: In accordance with your recommendation, we have edited the manuscript.

Reviewer 4 Report

Comments and Suggestions for Authors

No comments

Author Response

No comments made